# Effect of a Telephone-Based Lifestyle Intervention on Weight, Body Composition, and Metabolic Biomarkers in Rural Ohio: Results from a Randomized Pilot Study

**DOI:** 10.3390/nu15183998

**Published:** 2023-09-15

**Authors:** Xiaochen Zhang, Victoria R. DeScenza, Zachary L. Chaplow, Megan Kilar, Jessica Bowman, Alex Buga, Madison L. Kackley, Abigail B. Shoben, Ashley S. Felix, Electra D. Paskett, Brian C. Focht

**Affiliations:** 1Division of Cancer Prevention and Control, Department of Internal Medicine, The Ohio State University, Columbus, OH 43210, USA; zhang.8303@buckeyemail.osu.edu (X.Z.); electra.paskett@osumc.edu (E.D.P.); 2Division of Epidemiology, College of Public Health, The Ohio State University, Columbus, OH 43210, USA; ashley.felix@osumc.edu; 3Public Health Sciences, Cancer Prevention Program, Fred Hutchinson Cancer Center, Seattle, WA 98109, USA; 4Kinesiology, College of Agriculture, Health, and Natural Resource, University of Connecticut, Storrs, CT 06268, USA; tori.descenza@uconn.edu; 5Kinesiology, Department of Human Sciences, The Ohio State University, Columbus, OH 43210, USA; chaplow.1@osu.edu (Z.L.C.); megan.kilar1@gmail.com (M.K.); jessica.bowman2@nationwidechildrens.org (J.B.); buga.1@buckeyemail.osu.edu (A.B.); kackley.19@osu.edu (M.L.K.); 6Division of Biostatistics, College of Public Health, The Ohio State University, Columbus, OH 43210, USA; shoben.1@osu.edu

**Keywords:** weight loss, obesity, rural, remote, health disparities, metabolic biomarkers

## Abstract

Rural residents experience higher rates of obesity, obesity-related chronic diseases, and poorer lifestyle. Promoting physical activity and healthy eating are critical for rural residents; however, lack of resources and access barriers limit the feasibility of in-person lifestyle interventions. There is a need to design and deliver remotely accessible lifestyle interventions in this population. This pilot study examined the effect of a telephone-based lifestyle intervention on weight, body composition, lipids, and inflammatory biomarkers among rural Ohio residents. Rural Ohio adults with overweight/obesity (n = 40) were 2:1 randomized to a 15-week telephone-based lifestyle intervention (n = 27) or control group (n = 13). The lifestyle intervention group received weekly telephone counseling sessions emphasizing healthy eating and increasing physical activity. The control group received educational brochures describing physical activity and dietary recommendations. Weight, body composition, fasting blood lipids, and inflammatory biomarkers were objectively measured at baseline and 15 weeks at local community centers (trial registration#: NCT05040152 at ClinicalTrial.gov). Linear mixed models were used to examine change over time by group. Participants were mostly female, with an average age of 49 years. Over the 15-week trial, the lifestyle intervention showed superior improvements in total cholesterol (∆ = −18.7 ± 7.8 mg/dL, *p* = 0.02) and LDL (∆ = −17.1 ± 8.1 mg/dL, *p* = 0.04) vs. control, whereas no significant between-group differences in weight, body composition, or inflammation were observed. Our findings suggest that a 15-week telephone-based lifestyle intervention may offer metabolic benefits that reduce disease risk in rural adults with obesity. Future large-scale studies are needed to determine the efficacy of remotely accessible lifestyle interventions in rural populations, with the goal of reducing obesity-related disparities.

## 1. Introduction

In the United States, about one in five people live in a designated rural area [1]. Adults living in rural communities exhibit elevated rates of obesity compared to urban residents at 39.6% and 33.4%, respectively [1,2]. In the state of Ohio, 23.7% of the population is rural, and 40.8% of rural Ohioans have obesity [3]. As adiposity is linked to the most pervasive chronic diseases and is a risk factor for many of the most prevalent cancer types [4,5,6,7,8,9], it is not surprising that cancer incidence and mortality are disproportionately higher in rural areas compared to non-rural [10]. In fact, although cancer incidence and mortality have been declining nationally, this trend has not been observed in rural populations for lung, prostate, breast, and colorectal cancers [10,11]. Consequently, it is essential to investigate how to engage rural communities to ameliorate these trends through innovative and evidence-based translational research efforts targeting modifiable determinants of lifestyle behavior change.

The factors contributing to rural health disparities are complex. Evidence suggests that individuals in rural areas engage in more disease-risk behaviors including physical inactivity, unhealthy diet, and lower rates of vaccination and cancer screening [12]. Social determinants of health such as access to quality health care and health literacy represent additional barriers to the successful self-management of one’s health in rural areas. Furthermore, access to in-person, community-based weight management programming remains limited. For example, the national Diabetes Prevention Program has been implemented across the U.S.; however, less than 15% of rural counties had access to the program, compared to 48% in urban counties [13].

Indeed, although the effectiveness of lifestyle weight management interventions has long been established, studies have primarily been conducted in controlled settings with participants recruited from urban areas. To date, few research studies have reported the effects of fully remote, telephone-based lifestyle weight management counseling interventions in rural-dwelling adults with overweight and obesity. Of the relevant extant trials, individual and group telephone counseling have been compared [14], as well as individual telephone versus group face-to-face counseling [15], and group telephone versus individual and group face-to-face counseling implemented in primary care clinics [16]. Overall, considerable heterogeneity in scientific methodology limits the conclusions that can be made regarding the effect of telephone counseling alone on weight loss in rural adults. Alternatively, telephone counseling has been extensively evaluated in extended-care trials of weight loss maintenance with results suggesting that both individual and group-based telephone counseling may be efficacious methods of mitigating weight regain following primary intervention [15,17]. However, all of the trials to date have focused on the outcome of weight loss with few trials reporting changes in other clinically relevant outcomes such as glucose and lipids [14,15,16,17]. Critically, no other studies in overweight or obese rural adults have reported effects of telephone counseling intervention on change in body composition or inflammatory markers.

Given the disparities in overweight and obesity, significant barriers to accessing weight management programs, and dearth of scientific study in rural populations, it is important to examine alternative approaches to delivering and evaluating lifestyle weight management interventions. Telephone-based counseling is one method that may increase reach and scalability to better serve rural populations. Therefore, we conducted the Home-based Exercise in Rural Ohio (HERO) study. HERO was a randomized feasibility trial of a 15-week individual telephone-based behavioral counseling intervention versus an active control condition among overweight or obese adults living in rural Ohio counties. We used a point-of-care assessment approach to collect data on all study outcomes. In this paper, we report on the observed changes in body weight, body composition, blood lipids, and inflammatory markers. We hypothesized that the 15-week intervention would yield improvements in these outcomes. Results from this study will be used to subsequently optimize the intervention approach for development of a future large-scale comparative efficacy study.

## 2. Materials and Methods

### 2.1. Participants

The HERO study was a pilot study using a single-blind, two-group randomized controlled design. The objective of the pilot study was to establish the feasibility of conducting a 15-week lifestyle intervention by weekly phone calls for rural individuals who had overweight/obesity. Detailed study methods and feasibility were previously published [18]. Briefly, we used Facebook advertising, flyers, and a list of participants from a previous study to recruit participants. Participants were eligible if they: lived in a rural Ohio county; BMI ≥ 25 kg/m^2^; age between 20 and 64 years; not in any weight loss intervention or reach physical activity guidelines; able to walk two city blocks; and able to speak and read English. Individuals who had a prior cancer diagnosis or severe medical conditions that would preclude physical activity and dietary modifications, or who were pregnant or nursing, or were unable to give informed consent were excluded from the study. Individuals who reported health conditions were asked to obtain medical clearance from their doctors before the initial in-person visit.

After completing the online screening survey, eligible individuals were scheduled for the initial in-person visit at the nearest participating local community center for informed consent and the baseline assessment. Using computer-generated random allocation sequence, 40 participants were 2:1 randomized to (1) lifestyle intervention group or (2) active control group. The project manager was the only staff who had access to the group allocation, and who assigned participants according to their allocation. The study was approved by The Ohio State University Institutional Review Board (2021C0033) and registered on ClinicalTrial.gov as NCT05040152. All participants provided written informed consent prior to any study activities.

### 2.2. Intervention

Details of the theory-based lifestyle intervention, including the behavior change techniques and mechanisms of action can be found elsewhere [18]. Participants randomized to the lifestyle intervention group received health counseling sessions via phone calls (30–45 min/session, delivered by health coaches who were exercise physiologists or had experience with behavioral intervention) for 15 weeks. In each session, the health coach provided instructions for healthy eating and increasing physical activity. The targeted lifestyle modifications consisted of (1) a gradually progressed reduction in caloric intake (500–1000 kcal/day) to reach an individualized caloric goal (1200–1800 kcal/day); and (2) simultaneously increasing both aerobic physical activity and resistance exercise to meet the physical activity guidelines [19]. Health coaches also provided recommendations for participants to meet dietary guidelines through MyPlate (https://www.myplate.gov/) (accessed on 15 May 2021) [20]. The intervention strategies were tailored to meet the unique needs of rural participants by (1) utilizing local accessible resources to establish healthy behaviors for each participant, (2) providing practical techniques to address individual barriers, and (3) encouraging them to form a support network with family and friends.

Individuals in the active control group were provided brochures with the American Institute of Cancer Research (AICR) physical activity and dietary guidelines during the initial in-person visit [21]. Participants were encouraged to meet the physical activity and dietary guidelines. After completing the 15-week measures, the active control group received the identical lifestyle modification manual that had been initially provided to the lifestyle intervention group at baseline.

### 2.3. Measurement

Baseline and 15-week measurements were obtained by trained staff members who were blinded to group assignment. Demographic characteristics including age, sex, and race–ethnicity were self-reported. Physical activity was objectively measured using the activPAL^TM^ micro activity monitor (PAL Technologies Ltd., Glasgow, Scotland, UK). Participants were provided verbal and written instructions for wearing the activPAL^TM^. They were asked to wear the device 24 h per day for seven consecutive days after each in-person assessment.

### 2.4. Anthropometric Assessment and Body Composition

Body weight (kg) and height (cm) were assessed using a digital scale and scale-mounted stadiometer to calculate BMI (kg/m^2^). Body composition, including fat mass (lb), percent fat, lean mass (lb), percent lean mass, as well as visceral and subcutaneous fat mass (lb), was measured using a three-dimensional (3D) body scanner, Styku S100 (Styku L.L.C., Los Angeles, CA, USA). The Styku S100 scanner is a portable body scanner, which offers advantages for measuring body composition in rural and field study settings. Styku S100 scanner uses an infrared camera built into a stationary tower and a 360-degree rotating platform to capture body circumference and volumes. Using proprietary formulas, Styku software (version 4.3) calculates body fat percentage, lean mass percentage, and bone mass among other body composition measures, all of which are reliable measures compared with dual-energy X-ray absorptiometry [22].

### 2.5. Lipid Profile

Participants underwent fasting (≥8 h) blood sampling for lipid profiles. Prior to blood draw, urine specific gravity (USG) was assessed by a light refractometer (Reichert™, Buffalo, NY, USA). In instances where USG > 1.025, participants were instructed to consume a minimum of 250 mL of water until they achieved proper hydration. Lipid profiles including total cholesterol (TC), triglycerides (TG), high-density lipoprotein cholesterol (HDL-C), and low-density lipoprotein cholesterol (LDL-C) were acquired by fingerstick and analyzed using Cholestech LDX (Hayward, CA, USA). The Cholestech LDX has previously established good accuracy and precision in estimating whole-blood lipids and was used reliably as a point-of-care assessment for blood analyses in ambulatory settings [23,24].

### 2.6. Inflammatory Markers

Blood drawings (fasting ≥ 8 h) were performed by a trained phlebotomist after the finger stick, using antecubital puncture via butterfly needle (21G BD Eppendorf, Swedesboro, NJ, USA). Venous blood was collected in plasma EDTA (BD Vacutainer^®^, 10 mL, Swedesboro, NJ, USA), inverted 8 times, and centrifuged immediately after the blood draw (1500× *g*, 10 min, 4 °C). Plasma was aliquoted into 1000 μL screwcap cryovials and stored at −80 °C in a portable freezer until arrival to the laboratory center. *C*-reactive protein (CRP, mg/L), Interleukin-6 (IL-6; pg/mL), and Tumor Necrosis Factor alpha (TNF-α; pg/mL) were later assayed using the Meso QuickPlex SQ 120 sandwich immunoassay-VPlex Kit (Meso Scale Discovery, Rockville, MD, USA). For CRP, the lower limits of detection (LLD) was 1.33 mg/L; intra- and inter-assay coefficients were 2.9% and 7.9%, respectively. For IL-6, the LLD was 0.06 pg/mL; intra- and inter-assay coefficients were 4% and 6.4%, respectively. For TNF-α, the LLD was 0.04 pg/mL; intra- and inter-assay coefficients were 2.8% and 7.9%, respectively.

### 2.7. Statistical Analysis

For descriptive statistics at baseline, counts and proportions were reported for categorical variables and means ± standard deviations were reported for continuous variables. All inferential analyses were conducted on an intention-to-treat basis. Changes in body weight, body composition (fat mass, %fat, lean mass, %lean), lipids (TC, TG, HDL-C, LDL-C), and inflammatory markers (CRP, IL-6, TNF-α) were calculated from baseline to 15 weeks in the two groups using linear mixed-effects regression models. This statistical approach included all available data and accounted for the correlation between repeated measures [25]. Interaction terms between group and time were computed as fixed-effects in the regression model. We report estimates with least-squares mean (LS Mean) ± standard error (SE) based on this preliminary data to inform future studies. The level of significance was *p* < 0.05. The statistical analyses were performed from January to March 2023 using Stata version 17.0 (StataCorp, LLC., College Station, TX, USA).

## 3. Results

A total of 40 participants were recruited between November and December, 2021, and completed intervention and final data collection between March and May 2022 (Figure 1) [18]. Baseline characteristics of study participants are presented in Table 1. Briefly, 35 of 40 participants (88%) completed the 15-week online survey. Among participants from the lifestyle intervention group, 22 out of 27 (82%) completed the 15-week intervention. Body composition and biomarker data at 15-week follow-up were available on 19 of 27 (70.4%) participants in the lifestyle intervention group and 9 of 13 (69.2%) participants in the active control. No adverse events were reported.

Anthropometric and body composition outcomes are presented in Table 2. Compared with the control group, the lifestyle intervention group lost 2.84 ± 1.93 kg weight (Figure 2, *p* = 0.14) and reduced BMI by 0.85 ± 0.71 kg/m^2^ (*p* = 0.23). Specifically, participants in the lifestyle intervention lost 4.93 kg body weight (*p* < 0.001) and reduced BMI by 1.56 kg/m^2^ (*p* < 0.001), whereas participants in the control group did not have significant changes in body weight or BMI. Among participants who completed the 15-week body composition assessment, eight (42%) participants from the lifestyle intervention group attained clinically significant weight loss (≥5%), and 2 participants (22%) from the active control group attained ≥ 5% weight loss. There were no significant changes in fat mass, percent body fat, lean mass, percent lean, visceral fat mass, and subcutaneous fat mass in either group (Table 2).

Fasting lipids and inflammatory biomarker outcomes are presented in Table 3. Comparing the lifestyle intervention to the control group, we observed between-group differences in total cholesterol (Figure 3, −18.70 ± 7.84 mg/dL, *p* = 0.017) and LDL-C (−17.05 ± 8.12 mg/dL, *p* = 0.036). Although we observed a large magnitude of the difference in change between groups in triglycerides (−25.04 ± 23.88 mg/dL), it was not statistically significant (*p* = 0.29). Participants in the lifestyle intervention substantially lowered total cholesterol (−13.97 ± 4.71 mg/dL, *p* = 0.003), triglycerides (−44.95 ± 13.87 mg/dL, *p* = 0.001), and HDL-C (−5.50 ± 1.65 mg/dL, *p* = 0.001), while participants in the control group reduced HDL-C (−6.46 ± 2.19 mg/dL, *p* = 0.003) and increased LDL-C (16.27 ± 6.58 mg/dL, *p* = 0.013).

In terms of inflammatory biomarkers, no significant between-group differences in CRP, IL-6, or TNF-α were observed. No significant changes in IL-6 and TNF-α were observed in either group; however, CRP demonstrated a within-group reduction by 1.73 ± 0.89 mg/L (*p* = 0.052) in the lifestyle intervention and a significant reduction of 2.96 ± 1.20 mg/L (*p* = 0.014) in the control group.

## 4. Discussion

Overall, the main aim of the current study was to quantify the intervention effect of a telephone-based lifestyle intervention that combined exercise and dietary modification, in comparison to an active control group, among Ohio residents living in rural counties who had overweight or obesity. Our findings suggest that a 15-week telephone-based lifestyle intervention, compared to the health education control group, can provide weight loss benefit and favorable changes in fasting blood biomarkers, such as total cholesterol.

The telephone-based lifestyle intervention yielded significant weight loss in the intervention group, with an average of 4.9 kg lost between baseline and 15-week follow-up, which was greater than a recent meta-analysis that reported 1.8 kg weight loss in most rural weight loss trials [26]. Furthermore, 42% of participants from the lifestyle intervention group attained clinically significant weight loss (≥5%), which is aligned with results from other studies using similar intervention methods [26].

It is worth emphasizing that both groups had reductions in weight over the duration of the study, with the active control group experiencing a 2.1 kg loss of body weight at 15 weeks. The weight loss in the control group could be explained by them receiving the educational brochure which included physical activity and dietary guidelines. Two participants also acquired personal trainers to help them meet the exercise goals. Although the between-group differences in body weight were insignificant, the magnitude of the weight loss exhibited in our lifestyle intervention group is promising, especially over a short period of 15 weeks. In an urban study among women with higher risk of breast cancer, Hartman and colleagues found that self-monitoring combined with telephone-based counseling yielded significant weight loss, approximately 4.4 kg, over a 6-month study duration compared to 0.8 kg in the control group [27]. The HERO study utilized a similar approach to Hartman et al., but within a shortened timeframe in a rural setting, which demonstrates the potential for implementing a remotely accessible lifestyle intervention with a shorter duration in rural areas to achieve similar weight loss results.

Increased adiposity elevates the risk of cancer by the negative impacts on metabolic and inflammatory processes [6,7,8]. Recognizing the link between obesity and cancer risk, optimal weight loss would be a result of a decrease in fat mass and a concomitant increase or maintenance of lean mass. However, participants in our study did not yield significant changes in any body composition measures. The novel 3D body scanner showed validity and reliability in repeated measures [22]. However, the BMI of our study participants was greater than the samples in the validation studies, which may lead to insensitivity of detecting the changes in this study. A recently published pilot study examined the changes in body composition measured by bioelectrical impedance after a 26-week technology-based intervention with rural older adults [28]. The authors attributed a 10.1-pound average decrease in body weight to a decrease in fat mass and preservation of muscle mass through diet and exercise behavioral modifications. In addition to the smaller magnitude of weight loss, other factors may also explain the lack of changes in body composition in the current study, such as reduction in water weight and the actual magnitude of dietary modification, as well as engagement in the prescribed resistance training. Future studies should consider including various measures to examine how these factors influence changes in body composition from lifestyle interventions. It is also possible that longer intervention duration, including physical activity and dietary modification, may lead to substantial changes in body composition. Due to the short study period, participants were not able to experience significant change. Studies that include longer study duration with multiple assessments of body composition may help us understand when changes in body composition occur and how that aligns with changes in weight, lipids, and other disease factors.

Elevated levels of total cholesterol and triglycerides as well as low levels of HDL are associated with risk of cancers, suggesting that reducing total cholesterol and improving HDL may potentially reduce cancer risk [29]. Our study showed that total cholesterol and triglycerides were significantly reduced among the lifestyle intervention group, which highlights the potential of implementing a similar intervention approach to rural populations to reduce cancer risk.

Another important aspect of reducing obesity-related cancer risk is improved inflammatory biomarkers. Healthy lifestyle, including leisure time physical activity, healthy diet, and weight loss can improve energy balance and may reduce obesity-related inflammation, such as IL-6, TNF-α, and CRP [30,31,32]. Nevertheless, we found that both groups had improvement in CRP, but not in IL-6 or TNF-α. This might be explained by the fact that both groups experienced weight loss at 15 weeks, which may lead to a reduction in systematic inflammation. Further evaluation of the relationship between changes in physical activity, diet intake, and lipid and inflammatory profiles is warranted to help identify the optimal dose and intensity of lifestyle modifications in cancer prevention.

There are several notable strengths in our study. First, this was a randomized controlled pilot study, which by design, balanced the baseline confounding. We also kept staff blinded for the baseline and 15-week assessment to reduce measurement bias. Most importantly, we are, to date, the first study utilizing a portable 3D body scanner for body composition assessment in rural settings. Together with the portable lipid analyzer, we provided a point-of-care approach that was able to show participants their body composition and lipid results at the local study site. This approach may encourage participants to maintain a healthy lifestyle after study completion.

This study has several limitations. Due to the pilot study design, we conducted an exploratory analysis to quantify the intervention effect of a 15-week telephone-based intervention on weight, body composition, fasting blood lipids, and inflammatory biomarkers. We intend to use these findings as pilot data to support the design of a large-scale efficacy trial to demonstrate the metabolic benefits from weight loss interventions in rural populations with overweight/obesity. Therefore, the interpretation of these pilot study findings should be taken with caution. Additionally, participants who were interested and enrolled in the HERO study may not represent the general rural population with overweight/obesity. Moreover, although we established the feasibility of the lifestyle intervention in rural Ohio, we experienced some dropouts (17.5%). Compared to those who completed the study, those who dropped out had a higher BMI at baseline (Appendix A). Future research on the development of optimal retention strategies in weight loss interventions for this population is needed. Researchers may consider utilizing group teleconference to improve connection with participants and increase group engagement. To understand the effect of lifestyle intervention on systematic inflammation, future studies may consider extending the duration of the intervention, to allow sufficient time to evaluate the intervention efficacy on inflammation.

In conclusion, our pilot study offers important preliminary data and valuable experience in delivering a telephone-based lifestyle intervention for individuals with overweight/obesity in rural Ohio. Our findings demonstrate that a 15-week telephone-based lifestyle intervention may offer health benefits and reduce cancer-risk factors in rural populations. Future large-scale randomized trials are needed to examine the efficacy of remotely accessible weight loss programs in rural populations, with the goal to reduce obesity-related disparities.

## Figures and Tables

**Figure 1 nutrients-15-03998-f001:**
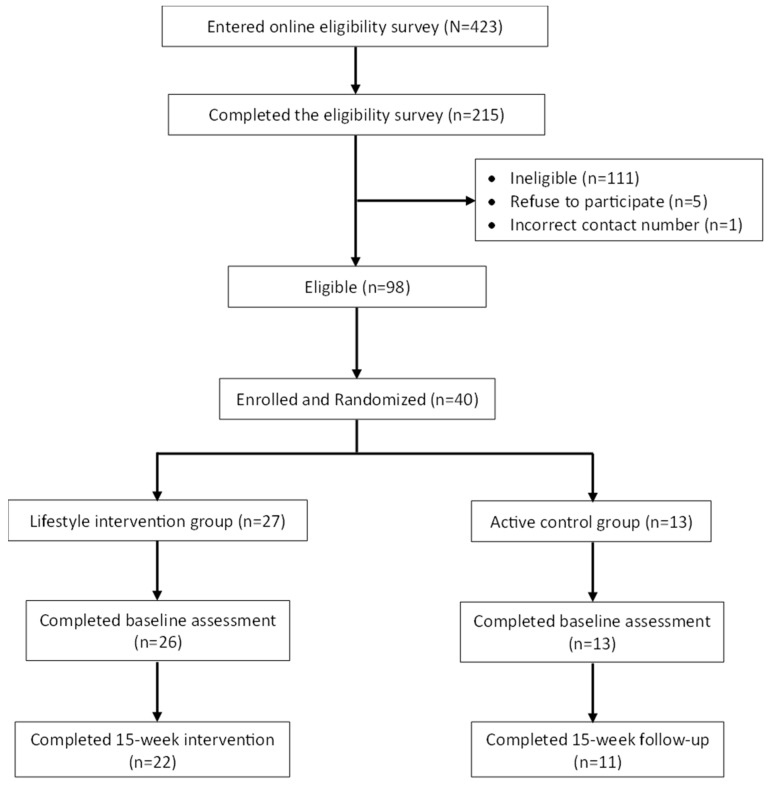
HERO Study CONSORT Diagram.

**Figure 2 nutrients-15-03998-f002:**
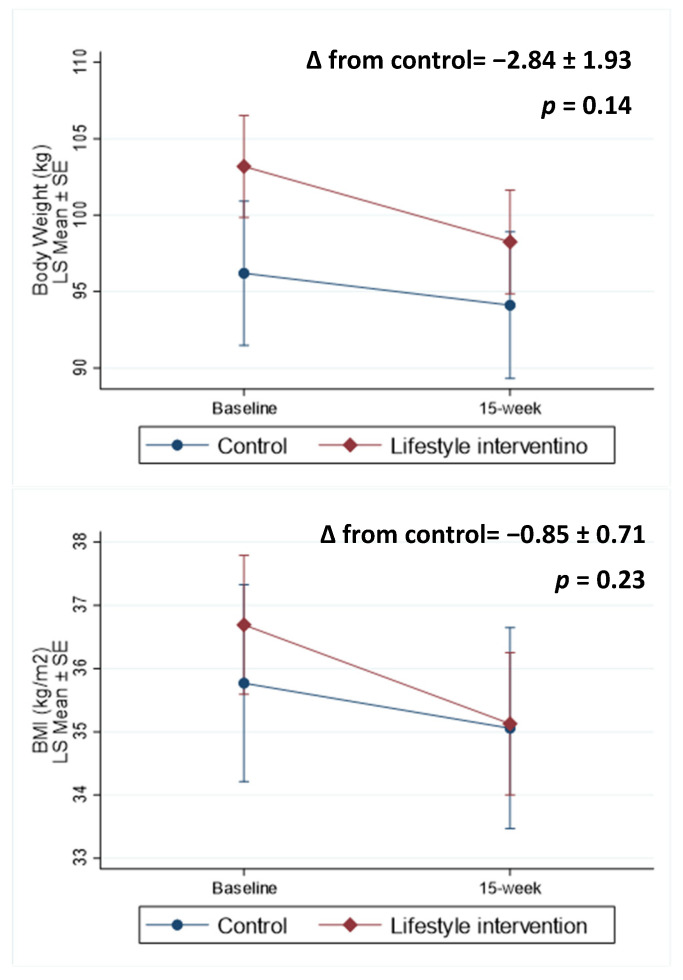
Body weight and BMI at baseline and change over 15 weeks. ∆: change.

**Figure 3 nutrients-15-03998-f003:**
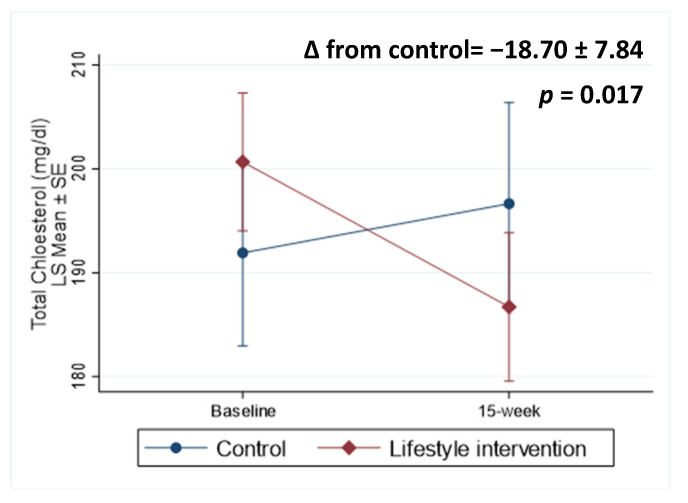
Lipid profiles at baseline and change over 15 weeks. ∆: change.

**Table 1 nutrients-15-03998-t001:** Baseline Characteristics of Participants by Study Group (N = 40).

	Lifestyle Intervention (n = 27)	Control (n = 13)
Age, years	49 ± 10	51 ± 9
Sex, n (%)		
Female, n (%)	24 (89%)	12 (92%)
Male, n (%)	3 (11%)	1 (8%)
BMI, kg/m^2^	36.7 ± 5.82	35.8 ± 5.49
Physical Activity, MET-h/day	33.3 ± 0.93	34.05 ± 1.29

**Table 2 nutrients-15-03998-t002:** Anthropometric and body composition outcomes at baseline and change over 15 weeks.

		Baseline	∆ Baseline to 15-Week		∆ from Control	
		Mean ± SD	LS Mean ± SE	*p*-Value	LS Mean ± SE	*p*-Value
**Anthropometric Outcomes**						
**Weight (kg)**						
	Control	96.20 ± 15.58	−2.09 ± 1.58	0.19		
	Intervention	103.20 ± 18.37	−4.93 ± 1.12	<0.001	−2.84 ± 1.93	0.14
**BMI (kg/m^2^)**						
	Control	35.77 ± 5.49	−0.71 ± 0.58	0.22		
	Intervention	36.69 ± 5.82	−1.56 ± 0.41	<0.001	−0.85 ± 0.71	0.23
**Waist Circumference (in)**						
	Control	38.18 ± 5.16	−0.24 ± 0.79	0.76		
	Intervention	38.98 ± 4.71	−0.25 ± 0.54	0.65	−0.007 ± 0.96	0.99
**Waist-to-hip ratio**						
	Control	0.90 ± 0.08	−0.012 ± 0.011	0.24		
	Intervention	0.87 ± 0.06	−0.001 ± 0.007	0.87	0.011 ± 0.013	0.39
**Body Composition Outcomes**						
**Fat (lb)**						
	Control	86.89 ± 19.23	0.23 ± 3.46	0.95		
	Intervention	92.02 ± 19.36	0.11 ± 2.38	0.97	0.12 ± 4.20	0.98
**% Fat**						
	Control	41.21 ± 4.50	−0.089 ± 0.84	0.92		
	Intervention	41.33 ± 5.06	0.27 ± 0.58	0.64	0.36 ± 1.02	0.73
**Lean (lb)**						
	Control	117.68 ± 18.36	0.71 ± 2.30	0.70		
	Intervention	125.41 ± 29.17	−1.80 ± 1.59	0.13	−2.50 ± 2.80	0.37
**% Lean**						
	Control	56.28 ± 4.26	0.084 ± 0.81	0.92		
	Intervention	56.13 ± 4.91	−0.27 ± 0.56	0.63	−0.36 ± 0.99	0.72
**Visceral fat mass (lb)**						
	Control	1.77 ± 0.63	−0.019 ± 0.100	0.85		
	Intervention	1.74 ± 0.44	0.039 ± 0.069	0.57	0.058 ± 0.12	0.63
**Subcutaneous fat mass (lb)**						
	Control	5.34 ± 1.26	0.033 ± 0.20	0.72		
	Intervention	5.63 ± 1.06	0.015 ± 0.14	0.65	−0.017 ± 0.24	0.94

∆: change.

**Table 3 nutrients-15-03998-t003:** Lipids and inflammatory biomarker outcomes at baseline and change over 15 weeks.

		Baseline	∆ Baseline to 15-Week		∆ from Control	
		Mean ± SD	LS Mean ± SE	*p*-Value	LS Mean ± SE	*p*-Value
** *Lipids* **						
**Total cholesterol (mg/dL)**						
	Control	191.92 ± 35.37	4.73 ± 6.27	0.45		
	Intervention	200.30 ± 32.72	−13.97 ± 4.71	0.003	−18.70 ± 7.84	0.017
**Triglycerides (mg/dL)**						
	Control	138.75 ± 59.49	−19.92 ± 19.43	0.31		
	Intervention	168.61 ± 95.50	−44.95 ± 13.87	0.001	−25.04 ± 23.88	0.29
**HDL-C (mg/dL)**						
	Control	56.08 ± 19.41	−6.46 ± 2.19	0.003		
	Intervention	50.57 ± 12.46	−5.50 ± 1.65	0.001	0.96 ± 2.74	0.73
**LDL-C (mg/dL)**						
	Control	115.75 ± 31.36	16.27 ± 6.58	0.013		
	Intervention	116.86 ± 30.58	−0.77 ± 4.76	0.87	−17.05 ± 8.12	0.036
** *Inflammatory Biomarkers* **						
**CRP (mg/L)**						
	Control	10.06 ± 6.42	−2.96 ± 1.20	0.014		
	Intervention	9.82 ± 7.31	−1.73 ± 0.89	0.052	1.23 ± 1.49	0.41
**IL-6 (pg/mL)**						
	Control	1.01 ± 0.55	−0.18 ± 0.15	0.25		
	Intervention	1.13 ± 0.50	−0.084 ± 0.11	0.46	0.09 ± 0.19	0.63
**TNF-α (pg/mL)**						
	Control	0.82 ± 0.56	−0.14 ± 0.11	0.20		
	Intervention	0.84 ± 0.44	−0.050 ± 0.081	0.54	0.09 ± 0.14	0.49

∆: change.

## Data Availability

The participants of this study did not give written consent for their data to be shared publicly, so due to the sensitive nature of the research, supporting data are not available.

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
