# Peer review of "Effect of a Telephone-Based Lifestyle Intervention on Weight, Body Composition, and Metabolic Biomarkers in Rural Ohio: Results from a Randomized Pilot Study"

_nutrients, 2023, doi:10.3390/nu15183998_

Round 1

Reviewer 1 Report

Summary

The study was conducted to evaluate a lifestyle intervention aimed at promoting weight loss and healthy behaviors. Participants were divided into two groups: the lifestyle intervention group, which received weekly telephone-based health counseling sessions for 15 weeks, and the active control group, which was provided with education brochures detailing physical activity and dietary guidelines. The lifestyle intervention emphasized a caloric intake restriction, an increase in physical activity to achieve 150-200 min/week of moderate-intensity aerobic activity, and resistance training. Health coaches provided guidance on dietary modifications and physical activity, tailoring strategies to meet the specific needs of rural participants. The active control group, post the 15-week period, received the same lifestyle modification manual given to the intervention group at the start. Measurements were taken at baseline and after 15 weeks by trained staff who were blinded to group assignments.

Overall, the lifestyle intervention group showed more pronounced improvements in weight and BMI compared to the control group. However, changes in other metrics like waist circumference and body composition were minimal and not significantly different between the two groups.

 Summary:

This study provides valuable data that display the opportunities by which weight management might be tackled in real life scenarios. The set up and methodology is decent (eligibility, screening,  randomization and intervention).

 Minor comments:

Eligible were N=98 subjects after screening, enrolled n=40.  Might there be a selection bias in this step?

It would have been nice to see this intervention then crossed over.

I find it important to report that LDL Chol was reduced in addition to neutral lipids (more than 10%).

Have the authors been able to measure meaningful TNFa level or were they at the level of detection (just curious) as this reviewer often sees changes in TNF in obesity cohorts (no inflammatory stimulation) but himself has never been able to successfully show meaningful TNFa level in non inflammatory conditions.

CRP rediction has been shown

What about blood count?

Have the authors been able to check for variance effects of medication or comorbidities/allergies

This reviewer would omit the reference to cancer risk factors (we talk here about a 15 week intervention). Systematic inflammation is hard to evaluate in this setting.

This reviewer would rather discuss opportunities to a) extend the study duration and b) find ways to also extend the intervention efficacy (groups?, field visits?) etc

Author Response

Minor comments:

Eligible were N=98 subjects after screening, enrolled n=40.  Might there be a selection bias in this step?

Authors response: We understand the concern about selection bias, however, we did not select who get to enrolled in the study. Eligible individuals were contacted on first come first serve basis. We contacted and scheduled the first in-person visit for 60 individuals, but only 41 showed up (one was not eligible after in-person assessment). We compared those who consented (n=41) and excluded (n=174) from the study in our feasibility paper (Plos one. 2023;18(3):e0282719). We did not see any difference in age, gender, and health conditions. Individuals who consented had a lower BMI than those were excluded, which was anticipated as individuals with a higher BMI may have functional limitation to participate in the unsupervised lifestyle intervention.

It would have been nice to see this intervention then crossed over.

Authors response: Agreed! We did not have enough funding that allow us to provide the intervention to the control group, or additional follow-up assessments.

I find it important to report that LDL Chol was reduced in addition to neutral lipids (more than 10%).

Authors response: Thank you.

Have the authors been able to measure meaningful TNFa level or were they at the level of detection (just curious) as this reviewer often sees changes in TNF in obesity cohorts (no inflammatory stimulation) but himself has never been able to successfully show meaningful TNFa level in non inflammatory conditions.

CRP rediction has been shown

Authors response: TNF-a were at the level of detection. Our understanding is that TNF-a is a serum inflammatory biomarker that is constitutively expressed by adipocytes, specifically the ones distributed as visceral fat. Its expression is in direct proportion to total fat mass. Changes in TNF-a have been observed in short- and long-term diet interventions, an effect that was dependent on changes in body fat mass. Our lack of significant TNF-a concentrations during the lifestyle intervention is unsurprising as participants did not meaningfully lose body fat (as measured by the 3D body scanner).

What about blood count?

Authors response: We did not measure blood count. This was one of my dissertation projects that I did not include all the measurements in the study. I will consider to include it in the future trials.

Have the authors been able to check for variance effects of medication or comorbidities/allergies

Authors response: Unfortunately, we did not. The objectives of the project were to establish the feasibility (published paper) and the preliminary efficacy of the intervention. We did not collect data on medication. In addition, the small sample size would not allow us to look into interaction/stratification.

This reviewer would omit the reference to cancer risk factors (we talk here about a 15 week intervention). Systematic inflammation is hard to evaluate in this setting.

Authors response: Thank you for confirming! We were excited to see the significant changes in lipids, and wanted to see whether inflammation could be improved. Maybe a longer intervention is needed to change the systematic inflammation.

This reviewer would rather discuss opportunities to a) extend the study duration and b) find ways to also extend the intervention efficacy (groups?, field visits?) etc

Authors response: Thank you for the comments. We revised the discussion (the end of the limitations) to suggest future study opportunities.

Reviewer 2 Report

Reviewer's comments on the manuscript entitled “Effect of a Telephone-based Lifestyle Intervention on Weight, Body Composition, and Metabolic Biomarkers in Rural Ohio: Results from a Randomized Pilot Study” (manuscript ID: nutrients-2586961). Some suggestions for the manuscript:

1. The manuscript must be adapted to the editorial requirements available on the journal's website.

2. I suggest that the authors start the abstract with 2-3 introductory sentences and only later present the purpose of the research.

3. I believe that the authors should work even more on the introduction, which would introduce the reader to the subject of the study. In the introduction, the authors place great emphasis on the impact of obesity on cancer and to a lesser extent on other lifestyle-related diseases. In my opinion, it is more important to review the literature to discuss the effects of various interventions on body weight in past studies and to show how the current study differs from those already published.

4. The authors should present criteria for exclusion from the research group.

5. How did the authors determine the minimum sample size for this study? What criteria did they use to include this number of respondents?

6. I suggest that the authors move Figure 1 to the methodological part. The sampling method is not the result of the study.

7. I suggest that the authors combine Table 2 and Table 3 into one and explain the abbreviations used below the table.

8. Is there a need to repeat the results in the text, a lot of them can be seen in the Tables?

9. Authors should first list the study's strengths and then its limitations.

Author Response

  1. The manuscript must be adapted to the editorial requirements available on the journal's website.

Authors response: revised as requested.

  1. I suggest that the authors start the abstract with 2-3 introductory sentences and only later present the purpose of the research.

Authors response: revised as requested.

  1. I believe that the authors should work even more on the introduction, which would introduce the reader to the subject of the study. In the introduction, the authors place great emphasis on the impact of obesity on cancer and to a lesser extent on other lifestyle-related diseases. In my opinion, it is more important to review the literature to discuss the effects of various interventions on body weight in past studies and to show how the current study differs from those already published.

Authors response: Thank you for the comments. We substantially changed the introduction section as requested.

  1. The authors should present criteria for exclusion from the research group.

Authors response: Thank you for the comment. We have the eligible criteria in line 146-150. For more detailed criteria, we chose not to include in this manuscript since we have published it in the feasibility paper (https://pubmed.ncbi.nlm.nih.gov/36928626/).

  1. How did the authors determine the minimum sample size for this study? What criteria did they use to include this number of respondents?

Authors response: The primary aim of HERO study was to determine the feasibility of the intervention. The sample size was calculated to establish the feasibility. Specifically, the feasibility of the study was defined as having at least 80% of participants complete the follow-up online surveys at week 15. The feasibility of the weight loss intervention was defined as the percentage of participants in the weight loss group who completed the 15-week intervention. With 27 participants in the weight loss group, if the true completion rate was 80%, we had 92.5% power to rule out the unacceptable 50% completion rate with 95% confidence (one-sided). We did not include this information in the current paper since it was not the focus of the paper, and it was published in the feasibility paper.

The funding of the project allowed us to include up to 45 individuals in total (30 in the intervention, 15 in the control). We scheduled 60 individuals for the first in-person visit (considering dropouts and no shows) and only 41 showed up (one was not eligible after assessment). That’s how we ended up with the 27 participants in the weight loss group and 13 in the control group.

  1. I suggest that the authors move Figure 1 to the methodological part. The sampling method is not the result of the study.

Authors response: We understand the concerns regarding the sampling. We agree that recruitment and enrolment should be included in the methods section. However, figure 1 included information about how many participants completed each assessment, which should be part of the results of the study.

  1. I suggest that the authors combine Table 2 and Table 3 into one and explain the abbreviations used below the table.

Authors response: Thank you for the comments. We opt not to change the tables since table 2 focused on the anthropometric outcomes, and table 3 focused on the biomarkers outcomes. Each of these tables included a lot of information. Separating the 2 tables will allow readers to follow easily.

  1. Is there a need to repeat the results in the text, a lot of them can be seen in the Tables?

Authors response: Thank you for the comments. We only stated the important findings in the text. There were a lot of information in the tables. We felt it was necessary to emphasize the meaningful changes over the 15-week intervention.

  1. Authors should first list the study's strengths and then its limitations.

Authors response: revised as requested.